

# *Minos*-mediated transgenesis in the pantry moth *Plodia interpunctella*

Donya N. Shodja, Luca Livraghi and Arnaud Martin

Department of Biological Sciences, George Washington University, Washington, District of Columbia, United States of America

## ABSTRACT

Transposon-mediated transgenesis has been widely used to study gene function in Lepidoptera, with *piggyBac* being the most commonly employed system. However, because the *piggyBac* transposase originates from a lepidopteran genome, it raises concerns about endogenous activation, remobilization, and silencing of transgenes, thus questioning its suitability as an optimal tool in Lepidoptera. As an alternative, we evaluated the dipteran-derived *Minos* transposase for stable germline transformation in the pantry moth, *Plodia interpunctella*. We injected syncytial embryos with transposase mRNA, along with donor plasmids encoding *3xP3::EGFP* and *3xP3::mCherry* markers of eye and glial tissues. Across multiple experiments, we found that $G_0$ injectees could transmit *Minos* transgenes through the germline even in the absence of visible marker expression in the soma, and that large mating pools of $G_0$ founders consistently produced transgenic offspring at efficiencies exceeding 10%. Using these methods, we generated transgenic lines with a dual expression plasmid, using *3xP3::mCherry* for driving red fluorescence in eyes and glial tissues, as well as the *Fibroin-L* promoter expressing the recently developed mBaoJin fluorescent protein in the silk glands. This demonstrated the feasibility of screening two pairs of promoter activity in tissues of interest. Collectively, these results—along with previous findings in the silkworm *Bombyx mori*—demonstrate that *Minos* achieves robust germline integration of transgenes in Lepidoptera, offering a valuable pathway to the genetic modification of species where the remobilization or suppression of *piggyBac* elements might be rampant.

## INTRODUCTION

There is growing interest in developing new laboratory model systems and methods for the study of functional genomics in Lepidoptera, an enormous insect lineage where the silkworm *Bombyx mori* has been a focal species for the study of gene function. The pantry moth *Plodia interpunctella* is a pest of stored food with life history traits that make it suitable for economical and long-term maintenance of genetic lines. We and others have previously established methods for non-homologous repair mediated Clustered Regularly Interspaced Short Palindromic Repeats (CRISPR) mutagenesis, CRISPR short edits based on homologous recombination, and *piggyBac*-mediated transgenesis in this organism (*Bossin et al., 2007*; *Heryanto, Mazo-Vargas & Martin, 2022*; *Heryanto et al., 2022*; *Shirk et al.,*

Corresponding authors
Donya N. Shodja,
donya.shodja@gwu.edu
Arnaud Martin,
arnaud@email.gwu.edu

*2023*). So far, we have failed to insert a transgenic cassette of several kilobases using CRISPR-related techniques based on homologous repair and non-homologous plasmid insertion. Transposon-based random insertion of transgenes thus remains a valuable method in *Plodia*, and we expect that the further optimization of such methods will potentiate fundamental studies of lepidopteran biology, notably because the reagents used in these experiments—the source of transposase mRNA and the transgene donor plasmids— are easy to transfer to new species without modification.

One potential limitation of transgenesis in Lepidoptera as currently implemented, however, is that the favored *piggyBac* transposase originated from a lepidopteran species, the cabbage looper *Trichoplusia ni* (*Fraser et al., 1985*). Several distant lepidopteran species are known to harbor nearly identical *piggyBac* elements in their genomes, raising concern that their endogenous activity could elicit remobilization of newly inserted cassettes (*Zimowska & Handler, 2006*; *Wu et al., 2008*; *Wang et al., 2010*). Consistent with these potential issues, two studies documented that *piggyBac* transgenes can be remobilized in *Bombyx* and hinder the interpretability of reporter assays (*Wang et al., 2015*; *Jia et al., 2021*). In addition, certain species could have evolved silencing mechanisms to limit *piggyBac* element invasion and transmission (*Zimowska & Handler, 2006*), as characterized in *Drosophila* flies that evolved resistance to P-element transposons (*Khurana et al., 2011*; *Teixeira et al., 2017*; *Moon et al., 2018*). Given these potential concerns, it is important to assess whether transposons originating from more distant lineages work with a similar efficiency in Lepidoptera.

Here we investigated the suitability of an alternative transposon-based method using the *Minos* transposase. *Minos* is a *Tc1/mariner* element derived from the dipteran *Drosophila hydei* that has has been widely used for transgenesis in insects (*Franz & Savakis, 1991*; *Loukeris et al., 1995*; *Loukeris et al., 1995*; *Kapetanaki et al., 2002*; *Pavlopoulos et al., 2004*; *Metaxakis et al., 2005*), and other invertebrates (*Zagoraiou et al., 2001*; *Hozumi et al., 2010*; *Kontarakis & Pavlopoulos, 2014*; *Jackson et al., 2024*; *Kuo et al., 2024*). Importantly, a previous study found that the *Minos* transposase can mediate efficient germline integration in *Bombyx* embryos when delivered as an *in vitro* transcribed mRNA (*Uchino et al., 2007*). We transferred these methods to *Plodia*, observed efficient germline transmission of two plasmids, validated the expression of three fluorescent markers, and overall, confirmed that *Minos* can mediate stable transgenesis in another lepidopteran laboratory system. We discuss strategies for the rapid generation of transgenic lines in *Plodia* and provide detailed methods to facilitate the use of this technique in other species.

## MATERIALS AND METHODS

Portions of this text were previously published as part of a preprint (*Shodja, Livraghi & Martin, 2025*). The Institutional Biosafety Committee at the George Washington University administered the approval of Plodia rearing and transgenesis protocols in the Martin lab (protocol #IBC-24-261).

## Plasmids

The *pMi[3xP3::EGFP-SV40]* donor plasmid and the *Minos* transposase transcription template plasmid *pBlueSKMimRNA* (Pavlopoulos et al., 2004) were sourced from Addgene (*RRID:Addgene_102540; RRID:Addgene_102535*). The *pMi[3xp3::mCherry-SV40, Plodia-FibL::mBaojin-P10]* plasmid was produced using Gibson Assembly, by replacing the *EGFP-SV40* section of *pMi[3xP3::EGFP-SV40]* with a synthetic (*mCherry-SV40, FibL::mBaoJin-P10)* gene fragment synthesized by Twist Bioscience, and is available from Addgene (*RRID:Addgene_240410*). The FibL promoter sequence corresponds to 1 kb of sequence upstream of the *P. interpunctella FibL* gene (https://www.ncbi.nlm.nih.gov/gene/?term=LOC128678647).

## Preparation of *Minos* transgenesis reagents

Plasmids were amplified in 50 mL liquid cultures of Luria-Bertani broth with 100 µg/mL Ampicillin, and plasmid DNA was purified with the ZymoPURE II Plasmid Midiprep Kit (Cat# D4200). Plasmids were eluted in 120 µL of elution buffer, yielding concentrations >4,800 ng/µL of plasmid DNA.

To produce *Minos* transposase mRNA, the *pBlueSKMimRNA* plasmid was linearized using the *Not*I-HF restriction enzyme (New England Biolabs, Ipswich, MA, USA), purified by gel extraction using the Zymoclean Gel DNA Recovery Kit (Zymo Research, Irvine, CA, USA), and further purified using the Zymo DNA Clean & Concentrator-25 kit with an elution volume of 13 µL of DNA elution buffer. Subsequently, the HiScribe T7 ARCA mRNA with tailing kit (New England Biolabs, Ipswich, MA, USA) was used for transcription and capping of *Minos* transposase mRNA, followed by purification using the MEGAClear Transcription Clean-Up Kit (Thermo Fisher Scientific, Waltham, MA, USA). The purified mRNA was eluted in 50 µL of the provided elution buffer, which was pre-heated to 65 °C and incubated on the filter column for 10 min in a 65 °C oven. After quantification with Nanodrop (Thermo Fisher Scientific, Waltham, MA, USA), the transposase mRNA was divided into >520 ng/µL one-time use aliquots and stored at −70 °C.

Injection mixes were freshly prepared before injection and consisted of 400 ng/µL *Minos* transposase mRNA, 200 ng/µL donor plasmid, and 0.05% Phenol Red (1:10 dilution of a 0.5% cell-culture grade Phenol Red solution; Sigma-Aldrich, St. Louis, MO, USA), brought to a 5 µL final volume with 1x *Bombyx* injection buffer (pH = 7.2, 0.5 mM $NaH_2PO_4$, 0.5 mM $Na_2HPO_4$, 5 mM KCl).

## *Plodia* strain

The white-eyed *wFog* laboratory strain of *Plodia interpunctella* was used to facilitate the screening of *3xP3*-driven fluorescence in pupal and adult eyes (Heryanto et al., 2022; Heryanto, Mazo-Vargas & Martin, 2022). This strain consists in the genome background of the *Dundee* strain (NCBI genome accession GCA_001368715.1), into which a *white* gene *c.737delC* mutation originating from the *Piw3* strain (NCBI genome accession: GCA_022985095.1) was introgressed during two generations of backcrossing and several rounds of sib-sib inbreeding (Heryanto et al., 2022).

### *Plodia* husbandry

The *wFog* strain and the transgenic broods were reared in the laboratory from egg to adulthood in an incubator set at 28 °C with 40–70% relative humidity and a 14:10 h light: dark cycle, following methods made available *via* the Open Science Framework repository (*Heryanto et al., 2025*). Briefly, egg laying was induced by $CO_2$ narcosis of adult stocks in an oviposition jar, and a weight boat containing 10–13 mg eggs was placed in a rearing container containing 45–50 g of wheat bran diet with 30% glycerol (Table S1). This life cycle spans 29 days from fertilization to a reproductively mature adult stock at 28 °C.

## Microinjection of *Plodia* syncytial embryos

The general procedure for microinjection as well as a list of suggested equipment is available online (*Heryanto, Mazo-Vargas & Martin, 2022*) and summarized below. *Minos* capped mRNA and the donor plasmid were mixed into a 5 µL aliquot (400:200 ng/µL mRNA:DNA) and kept on ice shortly before the procedure. Eggs were collected in an oviposition jar following $CO_2$ narcosis of a stock of *wFog* strain adults, aligned on parafilm, and microinjected between 20–50 min after egg laying (AEL) using borosilicate pulled glass needles and an air pressure microinjector. Eggs were then sealed with cyanoacrylate glue and left to develop at 25 °C or 28 °C in a closed tupperware with a wet paper towel.

## Fluorescent screening equipment

All life stages were screened under an Olympus SZX16 stereomicroscope equipped with a Lumencor SOLA Light Engine SM 5-LCR-VA lightsource, mounted to a trinocular tube connected to an Olympus DP73 digital color camera. Fluorophore expression was observed using Chroma Technology filter sets ET-EGFP 470/40× 510/20m green fluorescent protein (GFP) and AT-TRITC-REDSHFT 540/25X, 620/60m red fluorescent protein (RFP). Autofluorescence patterns are shown for uninjected individuals, using the GFP and RFP filter sets and exposure times consistent with the other images in this study (Fig. S1). It is worth adding that a yellow fluorescent protein filter set (*e.g.*, Chroma ET-EYFP 500/20× 535/30m) reduces cuticle autofluorescence and can be advantageous to use in addition to a GFP filter set for screening *EGFP* and *mBaoJin* expression. For small mosaic clones, or weak fluorescent signals, the YFP filter on its own may lead to false negatives.

## Screening of embryos and first instars larvae

To obtain synchronized embryos, mated adults are anesthetized by $CO_2$ narcosis and transferred to a glass jar with a mesh lid insert (*Heryanto et al., 2025*). $CO_2$ exposure triggers egg release in *Plodia*, and embryos are collected after 2–4 h, transferred to a glass petri dish, and incubation at 28 °C and saturating humidity (*e.g.*, in a sealed tupperware with a wet paper towel), until time of screening. The screening of embryos is most convenient when using crosses performed in relatively large pools, for example with more than 10 mated females. Screening of transgenic embryos is optimal at 68–74 h AEL if incubated at 28 °C, a time window where *3xP3*-driven fluorescence in presumptive glia and ocelli is discernible in the larval head and body. Earlier times of screening show weaker

fluorescence and confounding signals, likely due to the episomal expression of residual plasmids in large vitellophages (*Heryanto, Mazo-Vargas & Martin, 2022*). Hatching occurs at 78–82 h AEL in uninjected embryos and can be delayed by up to 12 h in $G_0$ embryos, due to injection stress.

## Screening of pupae and adults

The screening of pupae is ideal to verify the transgenic status of imagos in preparation of a new generation, and can also be the method of choice when screening offspring from small crosses (less than 10 females), where isolating synchronized embryos by $CO_2$ narcosis is not optimal. As pupae can be difficult to isolate from the larval diet, it is helpful to add strips of cardboard to larval containers that contain fifth instar larvae, as the corrugations within this material are the preferred pupation site that facilitates the collection of pupae (*Shim & Lee, 2015*). Pupae are then isolated from these lodges, as well as from those located within the diet, transferred to tissue culture dishes for screening and sex sorting, and subsequently combined according to the cross design in fresh containers to allow adult emergence and mating prior to initiating the next generation. Fluorescent and brightfield images of pupae were acquired using the same fluorescent microscopy set-up using the Olympus DP74 camera, and overlaid using the Screen blending mode in Adobe Illustrator. Brightfield images of transgenic $G_2$ adult eyes (Fig. 1F) were taken using a Keyence VHX-5000 digital microscope. Pupal stages (%, Figs. 1C, 2G) were estimated based on eye, wing, tarsal, and cuticle pigmentation (*Zimowska et al., 1991*). Adult moths can be screened in a petri dish held on ice to limit moth activity.

## Characterization of silk gland fluorescence

The *FibL::mBaojin* transgene drives strong silk gland fluorescence visible directly through the cuticle of all larval instars. The silk glands of a $G_2$ *FibL::mBaojin* transgenic fifth instar larva were dissected, fixed, and mounted following a previously described procedure (*Alqassar & Martin, 2025*), and imaged on an Olympus SZX16 fluorescence stereomicroscope, as well as on an Olympus FV1200 confocal microscope mounted with a 20x objective.

# RESULTS

## Somatic and germline transformation of a test *Minos* donor plasmid

To test for the efficacy of *Minos*-mediated transgenesis in *Plodia* embryos, we initially microinjected 1,317 eggs between 20–50 mins AEL, with the *pMi[3xP3::EGFP-SV40]* test donor plasmid (*Pavlopoulos et al., 2004*). This injection time window, as well as the microinjection technique and rearing conditions were comparable to our previous experiments using the *Hyperactive piggyBac* transposase (*HyPBase*) in *Plodia* (*Heryanto, Mazo-Vargas & Martin, 2022*). The injection mix consisted of capped poly-A tailed *Minos* mRNA, mixed with a donor plasmid for the integration of the *3xP3::EGFP* transgenesis marker cassette.

We conducted three injection trials using the *pMi[3xP3::EGFP-SV40]* plasmid. All three attempts resulted in detectable activity in a portion of $G_0$ embryos and pupae. Embryonic

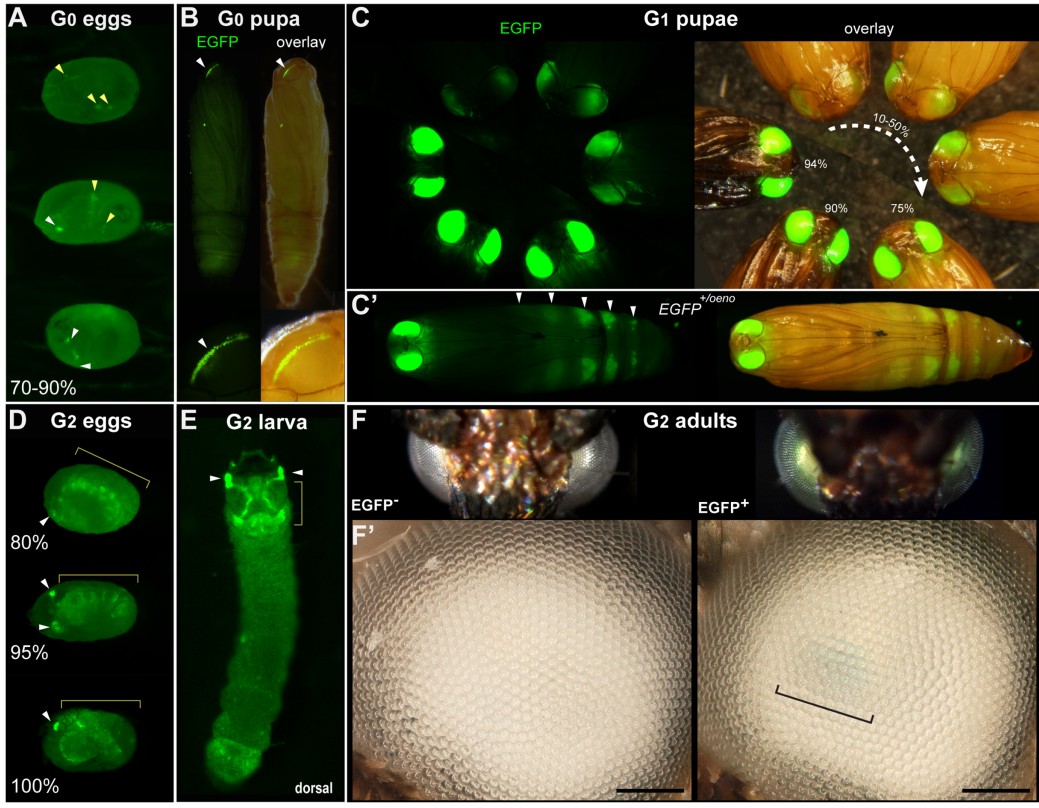

**Figure 1 Transgenic *Plodia* carrying the *3xP3::EGFP* cassette *via Minos*-mediated transgenesis.**
(A) $G_0$ eggs from injections of the donor plasmid *pMi[3xP3::EGFP]* and *Minos* transposase mRNA. Three examples of 3xP3 activity are shown at 70–90% of embryonic development. Effects are distinct from uninjected controls (Fig. S1). (B) Example of a mosaic *3xP3::EGFP* expression in a $G_0$ pupal eye. Right panels depict an overlay of EGFP and brightfield images taken separately. (C) Non-mosaic expression of EGFP in $G_1$ pupal eyes. EGFP intensity increases with the developmental stage of the pupae. Pupal stages (%) were estimated based on eye, wing, tarsal, and cuticle pigmentation (*Zimowska et al., 1991*). (C') Expression of the *EGFP +/oeno* allele, characterized by ectopic signal in abdominal segments (white arrowheads) in addition to the eyes. This ectopic activity marks oenocytes (Fig. S2) and was stable across subsequent generations. (D) $G_2$ embryos show expected *3xP3::EGFP* activity in the developing ocelli and presumptive glia. (E) *pMi[3xP3::EGFP]* expression in a first instar $G_2$ larva shows strong activity in head glia and lateral ocelli. (F–F') Adult $G_2$ individuals positive for *3xP3::EGFP* have green eyes under normal white light, due to strong EGFP expression. Comparison of an $G_2$ adult negative (left) and positive (right) for *3xP3::EGFP* under white light illumination and without fluorescence filters (F': magnified, lateral views of corresponding eyes). Bracket: visible green tint in inner retina. White arrowheads (A–B, D–E): fluorescent eye/ocelli (white). Yellow arrowheads and brackets (A–B, D–E): fluorescent glial tissues. Scale bars: F' = 100 μm.

activity of *3xP3::EGFP*, observed as of bright fluorescent puncta in the presumptive glia and larval ocelli (*Thomas et al., 2002*; *Bossin et al., 2007*; *Heryanto, Mazo-Vargas & Martin, 2022*; *Pearce et al., 2024*), was respectively detected in 10.3%, 7.8% and 4.3% of $G_0$ embryos in Experiments #1, #2 and #3 (Fig. 1A, Table 1). Pupal activity of *3xP3::EGFP*, consisting of bright fluorescent stripes of expression in the pupal eyes of the *wFog* white-eyed strain used here (*Heryanto, Mazo-Vargas & Martin, 2022*), was observed in 7.1%, 15.7%, and 16.7% of

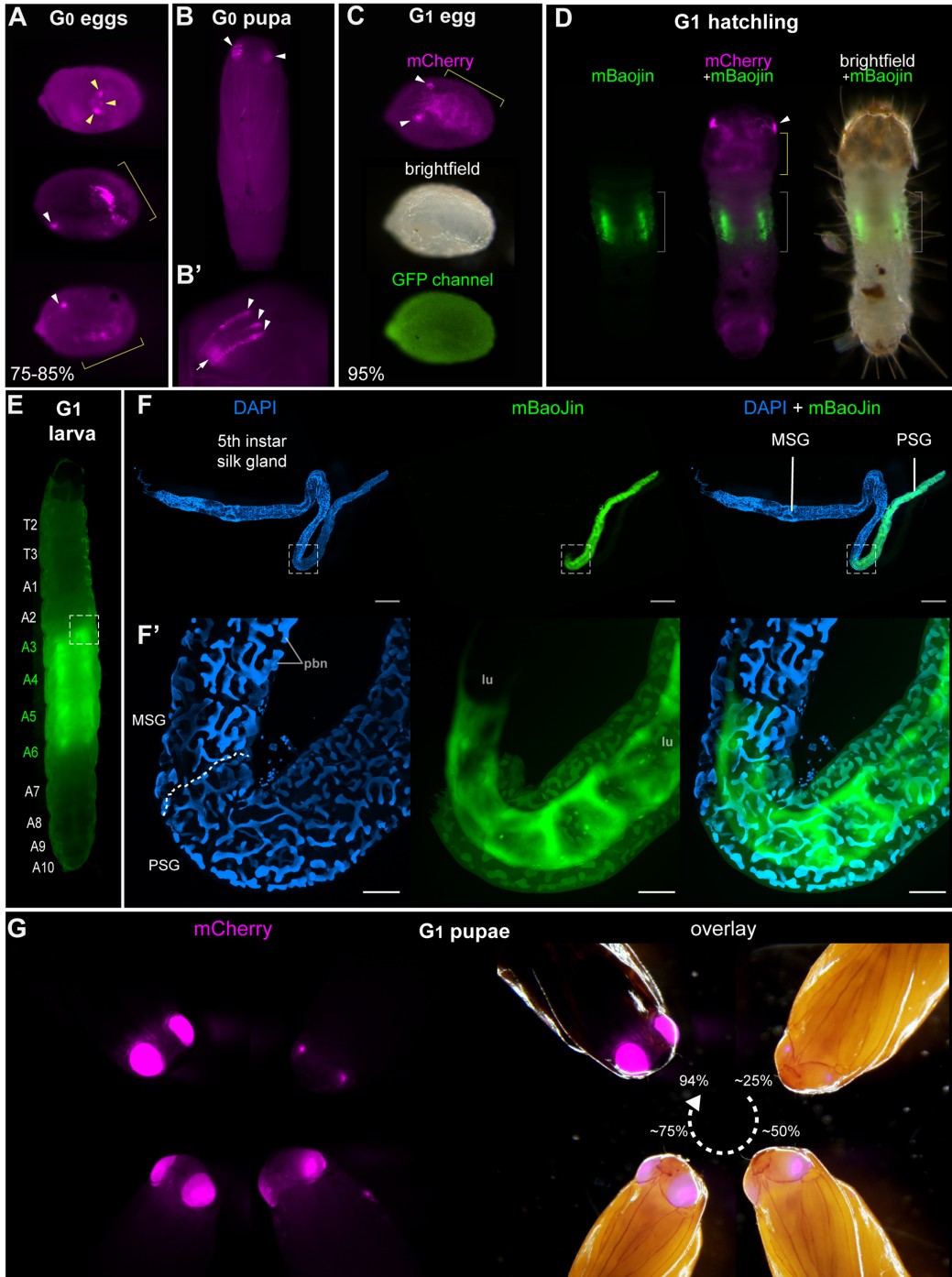

**Figure 2** **Tissue-specific activities of the *[3xP3::mCherry, FibL::mBaojin]* transgene following *Minos*-mediated integration.** (A) Mosaic expression of *3xP3::mCherry* in $G_0$ embryos, following injections of the donor plasmid *pMi[3xP3::mCherry, FibL::mBaojin]* and *Minos* transposase mRNA. Three examples of *3xP3* activity are shown at 75-85% of embryonic development. Arrowheads indicate *3xP3* specific signals in glia and ocelli. (B–B') Example of a mosaic *[3xP3::mCherry, FibL::mBaojin]* expression in a $G_0$ pupal eye. Arrow (B'): deeper non-retinal expression of *3xP3* in the optic lobe. (C) Dorsal views a $G_1$ embryo (95% stage), showing expected *3xP3::mCherry* activity in the developing ocelli and presumptive glia. Brightfield, and autofluorescence in the GFP channel are added underneath. No mBaoJin is detectable in embryos. (D) Aspect of *FibL::mBaoJin* green fluorescence in silk glands, directly visible
**Figure 2 (continued)**
through the dorsal epidermis of a $G_1[mCherry+]$ first instar larva. (E) *FibL::mBaoJin* expression in a $G_1$ fifth instar larva in segments $A_3$-$A_5$. (F-F') Dissected silk gland of a $G_1$ fifth instar larva expressing *FibL:: mBaojin* in the Posterior Silk Gland (green). F' panels show confocal stacks of the curved region around the MSG-PSG boundary (dotted line). DAPI counterstaining (blue) highlights polyploid branched nuclei (pbn), in the layer of large cells that surround the gland lumen (lu). The mBaojin signal is strongest in the PSG lumen and extends into the MSG lumen. (G–G') Expression of *3xP3::mCherry* in $G_1$ pupal eyes, increasing with the developmental stage of the pupae (shown as %). White arrowheads (A–D): fluorescent eyes/ocelli. Yellow arrowheads and brackets (A, C–D): fluorescent glia. Scale bars: F = 500 μm; F' = 100 μm.

**Table 1  Number of injected embryos and somatic transformation rates among $G_0$ individuals from Experiments #1–5.**

| Plasmid | Experiment (injection session) | G0 embryos | | | G0 pupae | | |
|---|---|---|---|---|---|---|---|
| | | Injected | *3xP3* somatic transf. | % somatic transf. | Total | *3xP3* somatic transf. | % somatic transf. |
| *pMi[3xP3::EGFP]* | **#1** | 341 | 35 | 10.3% | 14 | 1 | 7.1% |
| | **#2** | 696 | 54 | 7.8% | 51 | 8 | 15.7% |
| | **#3** | 280 | 12 | 4.3% | 24 | 4 | 16.7% |
| | *Sub-total* | *1,317* | *101* | *7.7%* | *89* | *13* | *14.6%* |
| *pMI[3xP3::mCherry, FibL:: mBaoJin]* | **#4** | 456 | 13 | 2.9% | 77 | 1 | 1.3% |
| | **#5** | 396 | 82 | 20.7% | 32 | 5 | 15.6% |
| | *Sub-total* | *852* | *95* | *11.2%* | *109* | *6* | *5.5%* |
| | **Total** | **2,169** | **196** | **9.0%** | **198** | **19** | **9.6%** |

**Note:**
transf, transformants with detectable *3xP3* activity.

pupae in Experiments #1, #2 and #3 (Fig. 1B, Table 1). Overall, these data indicate that *Minos* transposase can drive transgene insertion into the genome of somatic cells.

Next, we assessed whether *Minos* could efficiently transpose the test insert into the germline, and enable its transmission over subsequent generations. Following Experiment #1, we in-crossed all the 14 $G_0$ adults by pooling them for random mating and rearing in a single container. We recovered 250 $G_1$ pupae, 23.6% of which showed transgene activity (Fig. 1C, Table 2). Following Experiment #2, we similarly in-crossed 41 $G_0$ adults, but then directly screened embryos instead of pupae. This in-cross generated 470 $G_1$ embryos, of which 11.5% were transgenic (Table 2). It is worth noting that in these two crosses, 13 out of 14 of the $G_0$ founders from Experiment #1, and all of the $G_0$ founders from Experiment #2, lacked a detectable somatic activity of *3xP3::EGFP* at the pupal stage. For comparison, we out-crossed 5 EGFP$^+$ $G_0$ males from Experiment #2 to 10 females from the uninjected *wFog* strain, and obtained 10.6% transgenic embryos out of 113 $G_1$ eggs. In summary, both $G_0$ negative and positive individuals can be used as founders for germline transmission of *Minos* transgenes, as somatic transformants do not necessarily reflect whether germline transformation has occurred.

**Table 2  Cross-design and germline transmission of the *[3xP3::EGFP]* transgene.**

| Experiment | Gen. | Cross design | Embryos | | Pupae | | | Chi-2: 50% ratio, single insert |
|---|---|---|---|---|---|---|---|---|
| | | | Total | EGFP+ ocelli & glia | Total screened | EGFP+ eyes | EGFP+ oenocytes | |
| *pMi[3xP3:: EGFP]* **Experiment #1** | G1 | Random in-cross of 14 G0 founders (1 EGFP+, 13 EGFP−, sexes n.d.) | – | – | 250 | 59 (23.6%) | 10 (4%) | n.a. |
| | G2 | 4 ♀ wFog × **1 G1 [EGFP+]** ♂ | – | – | 24 | 17 (70.8%) | 0 | **n.s.** |
| | | **1 G1 [EGFP+]** ♀ × 4 G1 [EGFP−] ♂ | – | – | 104 | 70 (67.3%) | 0 | ***p < 0.01*** |
| | | 4 G1 [EGFP−] ♀ × **1 G1 [EGFP+/oeno]** ♂ | – | – | 80 | 43 (53.7%) | 43 (53.7%) | **n.s.** |
| | | 4 G1 [EGFP−] ♀ × **1 G1 [EGFP+/oeno]** ♂ | – | – | 75 | 40 (53.3%) | n.d. (<40) | **n.s.** |
| | G3 | **1 G2 [EGFP−]** ♀ × 4 wFog ♂ | – | – | 117 | 60 (51.3%) | 0 | **n.s.** |
| | | **1 G2 [EGFP+/oeno]** ♀ × 4 wFog ♂ | – | – | 139 | 64 (46.0%) | 64 (46.0%) | **n.s.** |
| | G4 | random in-cross of 60 **G3 [EGFP+]** | – | – | 30* | 30 (100%) | 0 | n.a. |
| | | random in-cross of 64 **G3 [EGFP+/oeno]** | – | – | 70* | 70 (100%) | 70 (100%) | n.a. |
| *pMi[3xP3:: EGFP]* **Experiment #2** | G1 | random in-cross of 41 G0 founders: 29 EGFP− ♀ x 12 EGFP− ♂ | 470 | 54 (11.5%) | × | – | – | n.a. |
| | | 5 G0 [EGFP+] ♂ x 10 ♀ wFog | 113 | 12 (10.6%) | × | – | – | n.a. |

Notes:
Only Experiments #1–2 are shown here, as $G_0$ injectees from Experiment #3 were not crossed to a next generation. The last column indicates the *p*-value of Chi² tests for difference to a 50% distribution. Results are in bold for back-crosses to wFog, where a 50% segregation is expected from a single-insertion in the transgenic donors.
Gen, generation; n.a, not applicable (a 50% distribution is not expected in these crosses); n.d., not determined; n.s., not significant ($p > 0.05$).
Asterisk, number indicates cases where only a fraction of individuals from a line were checked for fluorescence.
×, these experiments were stopped at the embryonic stage.

## Stable transgene transmission over several generations

Further examination of the germline-integrate *3xP3::EGFP* transgene activity revealed fluorescence patterns identical to the ones previously reported with *piggyBac*-mediated integration (*Heryanto, Mazo-Vargas & Martin, 2022*), consistent with glial expression in embryos, and ocelli expression in larvae (Figs. 1D, 1E, Video S1). In the depigmented eyes from the *wFog* strain, the retinal expression of EGFP was strong enough to reveal a green tint of the eyes, visible without fluorescent microscopy equipment (Fig. 1F).

In addition, out of the 59 $G_1$ pupae with eye fluorescence generated from Experiment #1, 10 showed an additional, unexpected fluorescence in pupal oenocytes (*Stendell, 1912*; *Shirk & Zimowska, 1997*; *Makki, Cinnamon & Gould, 2014*), visible as ventral metameric belts in the second to seventh abdominal segments (Fig. 1C', Fig. S2). This activity is likely due to an identical insertion shared between these 11 siblings. Below we refer to this randomly captured oenocyte activity as coming from a single *EGFP*[+/oeno] allele, and dub the transgene alleles that lack it *EGFP*[+]. To further test the transmissibility of these $G_1$ insertion alleles, we generated four crosses each consisting of a single *EGFP*[+] or EGFP[+/oeno] $G_1$ founder mixed with four non-transgenic individuals from the opposite sex. These crosses generated 60.2% transgenic $G_2$ offspring individuals on average (Table 2), including transmission of the *EGFP*[+/oeno] allele in the two corresponding crosses.

We then continued these experiments over the $G_3$ and $G_4$ generations to further assess transgene stability (Table 2). To obtain $G_3$ offspring, we performed two backcrosses each consisting of one positive $G_2$ female mixed with four non-transgenic white-eyed males,

**Table 3 Cross-design and germline transmission of the *[3xP3::mCherry, FibL::mBaoJin]* transgene.**

| Experiment | Gen. | Cross design | Embryos Total | mCher+ | Larvae Total | mCher+ | mBaoJ+ | Chi-2: 50% ratio, single insert |
|---|---|---|---|---|---|---|---|---|
| *pMi[3xP3::mCherry, FibL::mBaoJin]* **Experiment #4** | G1 | Random in-cross of 77 G0 founders: 1 mCherry+, 76 mCherry−, sexes n.d. | – | – | n.d. (>100) | 22 | 22 | n.a |
| | G2 | 4 wFog ♀ × 1 **G1 [mCherry+]** ♂ | – | – | 0 | – | – | |
| | | 4 wFog ♀ × 1 **G1 [mCherry+]** ♂ | – | – | 145 | 69 (47.5%) | 69 (47.5%) | **n.s.** |
| | | 1 **G1 [mCherry+]** ♀ × 4 wFog ♂ | – | – | 242 | 114 (47.1%) | 114 (47.1%) | **n.s.** |
| | | 1 **G1 [mCherry +]** ♀ × 4 wFog ♂ | – | – | 195 | 94 (48.2%) | 94 (48.2%) | **n.s.** |
| | | 6 **G1 [mCherry+]** ♀ × 8 **G1 [mCherry+]** ♂ | | | 74* | 65 (87.8%) | 65 (87.8%) | n.a |
| *pMi[3xP3::mCherry, FibL::mBaoJin]* **Experiment #5** | G1 | Random in-cross of 27 G0 founders: 13 [mCherry−] ♀ × 14 [mCherry−] ♂ | 518 | 51 (9.8%) | × | – | – | n.a |
| | | 4 wFog ♀ × 1 **[mCherry+]** ♂ | 168 | 0 (0%) | – | – | – | – |
| | | 4 **[mCherry+]** ♀ × 7 wFog ♂ | 160 | 0 (0%) | – | – | – | – |

Notes:
The last column indicates the *p*-value of Chi-square tests of difference to a 50% distribution. Results are in bold for back-crosses to wFog, where a 50% segregation is expected from a single-insertion in the transgenic donors.
Gen., generation; mCher[+], mCherry-positive; mBaoJ[+], mBaoJin-positive; n.a., not applicable (a 50% distribution is not expected in these crosses); n.d., not determined; n.s., not significant ($p > 0.05$).
Asterisk, number indicates cases where only a fraction of individuals from a line were checked for fluorescence.
×, these experiments were stopped at the embryonic stage.

yielding 51.3% [*EGFP*[+]] $G_3$ pupae in the first cross, and 46.0% [*EGFP*[+/oeno]] pupae in the second cross, each consistent with their donor parent carrying a single copy of a *EGFP*[+] or *EGFP*[+/oeno] insertion allele (Chi-square test $p > 0.7$; no significant deviation from 50% transmission). Pooled in-crossing of positive $G_3$ individuals yielded 100% $G_4$ positive progeny in both cases, with [*EGFP*[+]] or [*EGFP*[+/oeno]] fluorescent expression patterns consistent with their $G_2$ donor grand-parent. The persistence of ectopic oenocyte activity across four generations suggests that the location of the expression cassette was stable and had not been subjected to re-transposition.

## The *FibL* promoter enables fluorescent marking of the silk glands

Next, to provide an alternative marker to eye/glia in future experiments, we designed a new donor plasmid aimed at labeling the silk glands. In *Bombyx*, the promoter of the *Fibroin-L* silk gene (*FibL*) drives fluorescent protein expression in the posterior region of the silk glands, enabling detection through the cuticle. This promoter has notably been used as a transgenesis marker in experiments where the labeling of neural tissues is unwanted (*Imamura et al., 2003*; *Fujiwara et al., 2014*). To implement a similar strategy, we cloned a 1 kb fragment of the *Plodia FibL* proximal promoter to drive the expression of *mBaoJin*—a recently developed monomeric fluorophore with remarkable brightness and photostability (*Zhang et al., 2024*) followed by a P10 3′UTR sequence (*Xu et al., 2022*). This cassette was cloned upstream of a *3xP3::mCherry* red fluorescence eye marker in order to monitor successful integrations.

Injections of the resulting *Minos* donor plasmids yielded positive hatchling larvae and pupae expressing distinguishable mosaic patches of *3xP3::mCherry* expression in the eye/glia (Figs. 2A, 2B). We conducted two independent injections trials using this plasmid, Experiments #4 and #5. Experiments #4 generated only one somatic transformant out of 77 pupae (Table 1), and random in-crossing of this large pool of founders successfully generated transgenic 22 transgenic $G_1$ larvae, out of more than a hundred screened larvae (we failed to record an exact number of screened larvae, and can not provide a germline transmission rate in this experiment). These $G_1$ individuals successfully transmitted transgenes into $G_2$ backcrosses in a pattern consistent with the segregation of single-copies (Table 3). In Experiment #5, five out of 32 $G_0$ pupae showed mosaic *3xP3::mCherry* expression (Table 1), but these somatic transformants failed to pass a transgene into a $G_1$ generation, confirming that somatic mosaicism in the pupal eyes is a poor indicator of germline transformation. In contrast, a pooled in-cross of the 17 remaining negative $G_0$ individuals yielded a high-rate of germline transmission, with 51 out 518 embryos showing strong *3xP3* activity (Fig. 2C, Table 3).

In $G_1$ larvae, we detected *FibL::mBaojin* activity in sections of two inner tubes on either side of the gut, spanning the A3-A6 segments, directly screenable through the cuticle (Figs. 2C, 2D, Videos S2). Dissected glands show that *FibL::mBaoJin* fluorescence is confined to the Posterior Silk Gland (PSG, Figs. 2F, 2F'). This compartment of the silk gland is specialized in the expression of the fibroin genes *FibL* and *FibH* (Figs. 2F, 2F'), and is thus consistent with the *FibL* promoter recapitulating its endogenous expression in *Plodia* (Alqassar et al., 2025). Notably, in $G_1$ larvae, *FibL* expression in the PSG was also visible when screening with an RFP filter (Video S3), suggesting that the *FibL* promoter may be interacting with *mCherry* in addition to *mBaojin*, which could be due to the absence of insulator sequences between the two expression cassettes. $G_1$ donor successfully established $G_2$ lines in controlled crosses, including in four backcrosses where segregation ratios indicate that a single-insertion allele is present. Even in these $G_2$ individuals carrying a single-copy of the transgene, fluorescence of both the *mCherry* and *FibL* markers provided extremely bright signals that enable fast screening (Videos S4, S5).

Overall, these data further confirm the efficiency of *Minos* transgenesis for a second donor plasmid, and the *FibL::mBaoJin* cassette provides a marker that is easily screenable in *Plodia* throughout the entire larval stage (Fig. 3A). The *FibL* promoter could be used in the future to drive the expression of other fluorescent markers or ectopic silk factors in the *Plodia* Posterior Silk Gland.

## DISCUSSION

### Recommendations for successful transgenesis in *Plodia* and beyond

The feasibility of transgenesis experiments depends not only on the germline integration rate of the transposase system, but also on other logistic criteria such as the injection, rearing and crossing efforts. Below, we elaborate on these aspects and provide experimental considerations with the overarching goal to facilitate implementation in *Plodia* and other lepidopteran organisms.

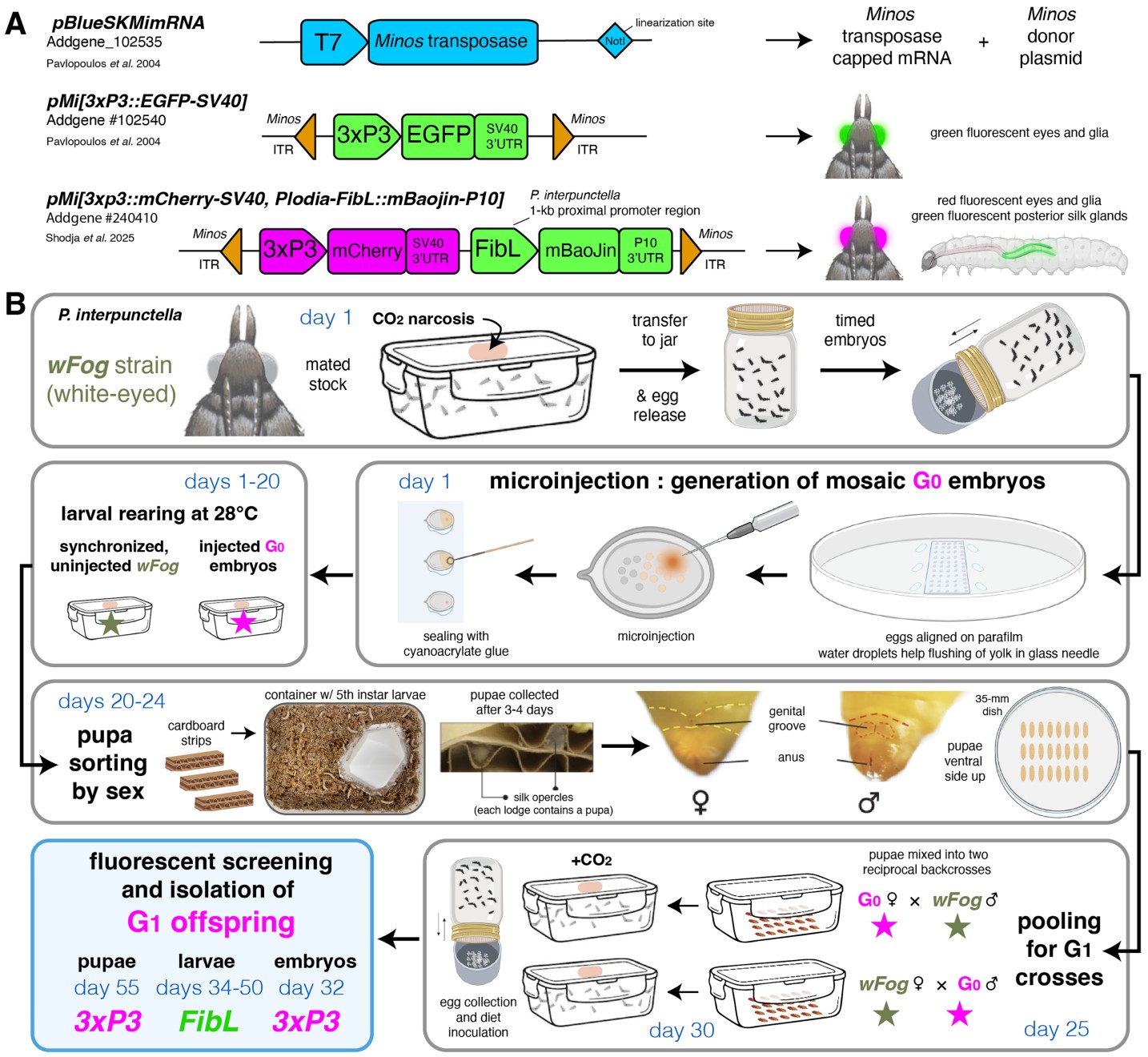

**Figure 3** *Minos* **transgenesis tools and procedure for the isolation of single-copy G₁ transgenic founders.** (A) Plasmids used in this study. (B) Suggested procedure for isolating single-copy transgenicheterozygotes at the G₁ generation, using reciprocal back-cross pools of G₀ and uninjected males and females. Alternatively, the G₀ injected stock can be randomly in-crossed if multiple transgene copies are acceptable or preferable, as performed in this study (Tables 2, 3). Chronology is indicated in days for a 28 °C rearing temperature. Full rearing procedures are available online (*Heryanto et al., 2025*).

We detected lower rates of G₀ mosaic fluorescence with *Minos* compared to previous *HyPBase* experiments, with 9.0% of the G₀ injected eggs and 9.6% of surviving pupae showing a *3xP3*-driven signal, compared to 22% of injected eggs and 32% of surviving

pupae with *HyPBase* (*Heryanto, Mazo-Vargas & Martin, 2022*). However, here we established that $G_0$ injectees devoid of visible somatic activity are effective germline carriers of the transgene. As a consequence, we propose that the random in-crossing of these negative $G_0$ injectees, performed in pools of adults, offers a practical route to the generation of transgenic lines. For example, in Experiment #1, 341 embryos were microinjected (an effort of only 3 h for two experimenters). In order to establish a $G_1$ line, all 14 surviving adults from the injected batch were transferred to a larval diet container and allowed to randomly mate. This approach generated a large proportion of $G_1$ individuals with *3xP3*-driven fluorescence, with 23.6% of 250 screened pupae showing EGFP expression in their eyes, which were then used to establish two transgenic lines (*EGFP*$^+$ and *EGFP*$^{+/oeno}$). In other words, we recommend in-crossing all $G_0$ individuals to ensure germline transmission and the generation of $G_1$ transformants. Indeed, crossing of small numbers of $G_0$ individuals can be time-consuming or yield unsuccessful matings, and selecting $G_0$ somatic transformants as founders entails the risk that the transgene has not been integrated into the germline. This approach is supported by our results from in-crossing all phenotypically $G_0$ negative individuals across several experiments, which all resulted in large $G_1$ cohorts with >10% transgenic individuals to select from to establish stable lines (Tables 2, 3). In contrast, backcrosses of positive $G_0$ individuals was successful in Experiment #2, but failed to produce any $G_1$ transformants in Experiment #5.

The pooled $G_0$ crossing scheme presents some caveats. First, not being able to trace how many individuals reproduced, we could not accurately measure germline integration rates as previously done in other insects, including our report of successful transgenesis in *Plodia* using the *Hyperactive piggyBac* transposase (*Gregory et al., 2016*; *Heryanto, Mazo-Vargas & Martin, 2022*). With the current data, it is unclear which of the *Minos* or *Hyperactive piggyBac* transposases drives the highest rates of germline transformation. Second, pooling $G_0$ siblings risks creating $G_1$ individuals inheriting several insertion sites. While having several insertions segregating in a line is not always an issue, *e.g.*, if a transgene is designed to fluorescently label structures for qualitative imaging, or for testing the ability of select promoters to drive tissue specific expression (*e.g.*, the *FibL* promoter), this could be a concern for quantitative or expression-reporter assays that may be influenced by positional effects. In such cases, we recommend sorting $G_0$ pupae by sex (*Heryanto et al., 2025*), and mix a pool of $G_0$ males to an equivalent number of uninjected females, and *vice-versa* (Fig. 3B). Generating a synchronized stock can facilitate this, simply by leaving a portion of the collected embryos uninjected at the beginning of the experiment.

Alternatively, if $G_1$ broods originate from an in-cross of pooled $G_0$ siblings, mixing single $G_1$ transgenic individuals with non-transgenic individuals should generate a majority of $G_2$ lines where a single transgene insert is segregating. It is worth noting that while we observed 50% segregation ratios consistent with single-inserts among our $G_2$-to-$G_4$ generations (Tables 2 and 3), we have not determined whether these lines include transcriptionally silent transgene copies, which might exist if *Minos*-mediated insertions can occur in heterochromatin regions. Inverse/splinkerette polymerase chain reaction (PCR) or sequencing techniques could be used in the future to map the position of inserts in the genome (*Potter & Luo, 2010*; *Pavlopoulos, 2011*; *Stern, 2016*).

### Flexible marker choice using *3xP3* and *FibL* drivers and various fluorophores

The *3xP3:EGFP* and *3xP3:mCherry* markers provide a reliable labeling on larval ocelli, glial tissue, and eyes across insects, including the hemimetabolous crickets and firebrats (*Gonzalez-Sqalli, Caron & Loppin, 2024*; *Inada, Ohde & Daimon, 2025*). This signal was strong enough to be screened throughout the cuticle throughout the entire larval stage of the *Plodia wFog* strain, in which a *white* gene null mutation not only results in white eyes but also in increase translucency of the larval cuticle (*Heryanto et al., 2022*). In species where opaque cuticle may block fluorescence, or where a white-eyed mutant line is not available, it is still be possible to screen for *3xP3* activity in the larval lateral ocelli (*Thomas et al., 2002*), or in early pupal eyes before melanization occurs (*Özsu et al., 2017*).

This said, these features make the *3xP3* constraining to use, and we expect *FibL*-driven fluorescence to provide an interesting alternative in some species, particularly those with a reasonably translucent or clear cuticle in the first larval instars. And not only *FibL* may be more convenient to screen, it may also be preferred for experimental reasons, particularly in neurogenetics studies where tagging neural tissues may be undesirable (*Fujiwara et al., 2014*).

Last, we tested mBaoJin for the first time in a lepidopteran insect, as a monomeric version of the StayGold fluorophores, which show improved photostability and brightness over existing green fluorescent proteins (*Zhang et al., 2024*). Its successful tagging of the silk gland makes it a promising tool as a fusion-protein or expression reporter for Lepidoptera *in-vivo* studies. A plasmid carrying *Minos* repeats, *3xP3:mCherry*, and the *Plodia*-derived *FibL:mBaoJin* marker is available on the Addgene repository for academic use (Addgene #240410).

## CONCLUSIONS

Our data, together with a previous report in *Bombyx* (*Uchino et al., 2007*), show that *Minos* transgenesis is applicable to lepidopteran systems, provided it is possible to inject early syncytial embryos and maintain relatively inbred lines. Given the risks of transgene remobilization and instability associated with *piggyBac* transposases of lepidopteran origin, we urge researchers interested in developing transgenesis in moths or butterflies to consider using *Minos*-based tools.

## ACKNOWLEDGEMENTS

We thank Jasmine Alqassar and Christa Heryanto for maintaining *Plodia* cultures Patricia Hernandez for providing access to microscopy equipment.

### Funding

This work was funded by the Human Frontiers Science Program grant RGP0029/2022 and the National Science Foundation grant MCB-2217159 to AM. The funders had no role in

study design, data collection and analysis, decision to publish, or preparation of the manuscript.

### Grant Disclosures
The following grant information was disclosed by the authors:
Human Frontiers Science Program: RGP0029/2022.
National Science Foundation: MCB-2217159.

### Competing Interests
The authors declare that they have no competing interests.

### Author Contributions
- Donya N. Shodja conceived and designed the experiments, performed the experiments, analyzed the data, prepared figures and/or tables, authored or reviewed drafts of the article, and approved the final draft.
- Luca Livraghi performed the experiments, prepared figures and/or tables, and approved the final draft.
- Arnaud Martin conceived and designed the experiments, performed the experiments, analyzed the data, prepared figures and/or tables, authored or reviewed drafts of the article, and approved the final draft.

### Ethics
The following information was supplied relating to ethical approvals (i.e., approving body and any reference numbers):

The generation and handling of recombinant DNA and transgenic insects were performed in BioSafety Level-1 (BSL-1) and Arthropod Containment Level-2 (ACL-2) conditions, as approved under biosafety protocol #IBC-24-261 at the GWU.

### Data Availability
The data is available in the figures and tables.

### Supplemental Information
Supplemental information for this article can be found online at http://dx.doi.org/10.7717/peerj.20249#supplemental-information.

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
