# Peer review of "Minos-mediated transgenesis in the pantry moth Plodia interpunctella"

_PeerJ, doi:10.7717/peerj.20249_

## Round 0.1 · original submission · Major Revisions

· Academic Editor

Major Revisions

Based on the reviewers’ comments, a major revision is recommended for the manuscript titled "Minos-mediated transgenesis in the pantry moth Plodia interpunctella" by Donya N. Shodja and Arnaud Martin. The current version presents a promising contribution by demonstrating Minos transposase-mediated transformation in a non-model lepidopteran species. However, significant improvements in experimental depth, data presentation, and contextual discussion are necessary before the manuscript can be considered for publication.

Major Revision Recommendation

Summary of Strengths:
The study explores an important question regarding stable transformation in Lepidoptera using Minos, a potentially more stable alternative to piggyBac.

Experimental procedures are well-documented and replicable.

Fluorescent imaging and general figure quality are strong.

The manuscript contributes to expanding the transgenesis toolkit for Plodia and other non-model insects.

Essential Revisions Required:
1. Experimental Replication and Extension:
Single Injection Limitation: All reviewers highlighted that the data are based on only one injection trial (341 embryos). This raises concerns about reproducibility and limits conclusions regarding transformation efficiency and survival.

Action: Conduct additional independent injection trials using both the current and alternative donor plasmids (e.g., additional markers).

Action: Provide replication data to assess reproducibility of Minos-mediated transformation.

2. Transgene Integration and Insertion Site Verification:
Insertion Site Mapping: As the genome of Plodia is available, at least one transgene integration site should be identified to confirm genomic insertion and support the claim of stable transformation.

Action: Perform inverse PCR or another suitable method to verify the genomic insertion site(s) of the transgene.

3. Missing Controls:
Negative Controls for Fluorescence: Reviewers requested inclusion of autofluorescence controls to help distinguish true EGFP expression.

Action: Include appropriate negative controls in fluorescence imaging (e.g., non-injected or mock-injected samples).

Injection Controls: Survival/mortality analysis lacks non-injected or control-injected embryos.

Action: Include and report survival rates for non-injected embryos and embryos injected with only one component (e.g., plasmid or mRNA) as baseline controls.

4. Analysis of Transmission and Inheritance Patterns:
G1 to G2 Transmission Anomaly: Some EGFP+ G1 crosses yielded >50% fluorescent offspring. This anomaly, and the absence of G2 transmission in some lines, raises questions about copy number, chromosomal location (sex vs. autosomes), and stability.

Action: Investigate transgene copy number (e.g., by qPCR or Southern blot) and evaluate sex-linkage by analyzing segregation patterns in reciprocal crosses.

Action: Extend crosses to G3 if data are available to assess transgene stability over generations.

5. Figure and Table Improvements:
Figure 1 Additions and Corrections:

Add brightfield overlay for Fig. 1B.

Remove or clarify the rightmost arrow in Fig. 1C’.

Table 1 Enhancements:

Include sex of G0 founders and mating configurations.

Report detailed survival rates and transgene inheritance in offspring from each cross.

Discuss potential reasons for differing progeny numbers and EGFP expression.

6. Broader Discussion and Contextualization:
Comparative Context with piggyBac:

Discuss Minos vs. piggyBac integration efficiency and remobilization using both literature and author-generated data.

Clarify the current status of previously created piggyBac Plodia lines and whether remobilization has been observed.

Hypothesis Justification: The rationale for choosing Minos based on potential stability should be better contextualized with supporting references and critical discussion of known remobilization events in Lepidoptera.

Germline Stability and Efficiency: Acknowledge that limited injection data prevents a full comparison of transposon efficiency/stability and reframe conclusions with more caution.

7. Raw Data Availability:
Transparency: Make raw data (e.g., fluorescence quantifications, raw counts of progeny, imaging files) available through supplementary materials or a public repository.

Recommendation Summary:
This manuscript presents valuable preliminary data on Minos-mediated transformation in Plodia interpunctella, but it requires significant revisions to meet the standards for publication. I recommend a major revision focusing on experimental replication, integration site validation, appropriate controls, and expanded analysis of transgene inheritance and stability. Enhancing the manuscript in these areas will significantly strengthen its impact and utility for the broader transgenesis community.

·

Basic reporting

In this manuscript, the authors study the applicability of Minos-mediated transgenesis in Plodia interpunctella. piggyBac-mediated transgenesis has been established in Lepidopterans, but has been shown to remobilize due to endogenous piggyBac-family transposase activity. To circumvent this problem, the authors attempt to establish Minos-mediated transgenesis in the pantry moth.

Experimental design

The authors provide clear and detailed directions, which, to this reviewer, are amply sufficient for other scientists to be able to repeat this protocol in their own labs as well as in other comparable lepidopterans.

Validity of the findings

The authors have convincingly shown that they can produce germline stable transgenesis, albeit at a lower transmission efficiency than the piggyBac mediated transgenesis.

Reviewer 2 ·

Basic reporting

Shodja and Arnaud Martin demonstrate the utility of the Minos transposase for achieving stable germline transformation in Plodia interpunctella. The article is clear, well-structured, and well-written. The introduction and background information are relevant, and the figures are professional, clean, and well-organized. However, the authors did not share the raw data. I commend them for making these findings readily available, as they contribute to expanding the toolkit for transgenesis in non-model lepidopteran species. I offer the following minor comments and questions, which the authors may consider if they find them useful for improving the manuscript.

I believe the article would benefit from the inclusion of additional data. The results presented are limited to a single injection trial (341 embryos) using one marker (3xP3-EGFP). In contrast, the authors have previously published work on P. interpunctella transgenesis using hyPBase, which included three different markers and multiple injection trials. Cloning those additional markers into the donor plasmid and conducting another round of injections seems like a feasible extension for this study. Could the authors consider providing further data, rather than limiting the current manuscript to the results of a single injection trial?

Experimental design

In general, the authors followed a rigorous and standardized protocol for insect transgenesis, presenting a well-defined research question, effective methodologies, and procedures that are replicable by other researchers.

The authors justify the use of the Minos transposon based on the hypothesis that it may be less prone to remobilization in lepidopteran genomes compared to piggyBac. However, this hypothesis is not directly tested in the study, nor is the experimental evidence cited to support it (Wang et al. 2015; Ji et al. 2021) further explored in the discussion. Could these be isolated cases of transposon remobilization? What is the current status of the hyPBase-derived Plodia transgenic lines previously generated by the authors? To date, is there any other empirical evidence supporting greater stability of Minos in this species? Including a paragraph or two on this topic would enhance the coherence and readability of the manuscript.

Additionally, I am curious as to why the authors did not include controls for embryo survival, such as non-injected embryos or embryos injected with only individual components of the injection mix. Such controls are essential to discern the intrinsic causes of post-injection mortality, including potential toxicity from plasmids or mRNA, or adverse environmental conditions. These data could also provide a valuable baseline for future comparisons, particularly if Minos-mediated transgenesis proves to be more stable than piggyBac-mediated approaches.

Validity of the findings

The findings presented are valid and merit publication. However, although the authors demonstrate successful transgenesis in Plodia interpunctella using Minos, the number of injection trials is insufficient to support formal comparisons of transgenesis efficiency between Minos and hyPBase, or to test any hypotheses related to transgene instability or remobilization. Did the authors generate stable lines as part of this study? While this may be a minor limitation, given that the manuscript primarily aims to present the efficient protocol for testing Minos-mediated transgenesis in P. interpunctella, it would strengthen the manuscript to acknowledge this directly in the discussion and to adopt a more cautious tone in certain conclusions.

Table 1 shows that although EGFP-positive G1 males and females were obtained, the transgene was not transmitted to the G2 generation. There are no results presented for the G3, but if such data now exist, it would be useful to know whether the same pattern persists. How do the authors explain this lack of transmission? Could it be due to variation in screening methods, inconsistencies in marker expression, or actual loss of the marker from the genome? A discussion of these possibilities would add valuable context and clarity to the results.

Additionally, I believe Table 1 could be improved to better communicate the underlying data. As a reader, it would be helpful to see more detailed information from the injection and crossing experiments—specifically, the sex of the G0 individuals and survival rates of the G2 progeny from each cross. For example, why does mating one EGFP-positive female with four EGFP-negative males apparently produce more pupae than the reciprocal cross (four EGFP-negative females with one EGFP-positive male)? How many females in each group actually mated? How many of the resulting eggs expressed EGFP? Including this level of detail, along with a brief analysis of mortality rates, would enhance the transparency and interpretability of the data. I encourage the authors to incorporate these points into the discussion.

Additional comments

The manuscript fits well within the journal’s aims and scope. I believe it is suitable for publication in its current form; however, as mentioned, there is always room for improvement. I would appreciate the authors’ responses to my comments though, both to continue the scientific dialogue and to satisfy my curiosity about why certain aspects were left out of the discussion.

Reviewer 3 ·

Basic reporting

This manuscript describes transgenesis of Plodia by tranposon Minos.
The fluorescent images of G0, G1, G2 generation indicate transgenesis by Minos. I suggest to add several experiments/data to strengthen this paper.

Experimental design

Since the data shown in this manuscript is derived from a single injection experiment,and other donor plasmids/transgenic markers has not been tested in this work,transgene insertion site should be investigated to support the detection of Minos-based transgenesis. Since genome sequence of Plodia has been sequenced, this could be achieved by inverse PCR.

Validity of the findings

I suggest adding negative controls to Fig1A-E, so that readers unfamiliar with Plodia can distinguish the weak EGFP signals from autofluorescence. For instance, negative control will be helpful to judge whether all of the fluorescent signals in the middle egg of Fig1A are from the 3XP3 EGFP transgene or not.

The fact that the proportion of EGFP+ individuals among the offspring resulting from crosses between G1 and white-eyed individuals exceeds 50% should be also discussed; the number of transgene insertions in the genome of EGFP+ G1 individuals and whether the transgene insertions are in sex chromosomes or autosomes should be examined.

From the results shown in this study, it is understandable that Minos would be the first choice when there is need to transform a line that already carries a transgene inserted by piggyBac.
However, since the injection experiment was performed only once, readers cannot judge to what extent Minos is comparable to piggyBac. Especially, it cannot be determined whether the low survival rate of the G0 generation is due to the use of Minos, or other factors.
Also it would be useful if the the germline integration rates for piggyBac (HyPBase) from previous work are shown in discussion.

Additional comments

1)The bright field photo should be added for Fig1B to show the shape of the pupa and the location of the GFP signal in the pupa.

2)The right most arrowhead in Fig1C' seems unecessary

---

## Round 0.2 · Minor Revisions

· Academic Editor

Minor Revisions

Dear Dr. Martin,

Please address the comments of reviewer #3

Best regards
Rodrigo

Reviewer 2 ·

Basic reporting

This new version of this article includes most of the recommendations I suggested. I feel it is ready to be published,

Experimental design

This new version of this article includes most of the recommendations I suggested. I feel it is ready to be published,

Validity of the findings

This new version of this article includes most of the recommendations I suggested. I feel it is ready to be published,

Additional comments

I think the authors did an excellent job with this revised version of the paper, incorporating new data, improving the figures and tables, and expanding the discussion. Congratulations!

Reviewer 3 ·

Basic reporting

no comment

Experimental design

no comment

Validity of the findings

no comment

Additional comments

The authors carefully addressed my comments/suggestions and the revised manuscript has been significantly improved.Therefore, I believe that the revised manuscript can be accepted after the following points.

1) The figure legend for Figure S1 is missing.In particular, an explanation is needed regarding the meaning of the arrowheads (fluorescent signals derived from transgenic markers?) and asterisks (individuals negative for transgenic signals) in the figure.
2) I assume the GFP and RFP images of Figure S1A were taken separately. Add a line between them to indicate this.

---

## Round 0.3 · accepted · Accept

· Academic Editor

Accept

Congratulations on the acceptance of your manuscript.

Reviewer 3 ·

Basic reporting

This new version of this article includes most of the recommendations I suggested. I feel it is ready to be published.

Experimental design

This new version of this article includes most of the recommendations I suggested. I feel it is ready to be published.

Validity of the findings

This new version of this article includes most of the recommendations I suggested. I feel it is ready to be published.